# ^18^F-Fluorodeoxyglucose PET/CT for Early Prediction of Outcomes in Patients with Advanced Lung Adenocarcinomas and EGFR Mutations Treated with First-Line EGFR-TKIs

**DOI:** 10.3390/cancers14061507

**Published:** 2022-03-15

**Authors:** Yu-Erh Huang, Ying-Huang Tsai, Yu-Jie Huang, Jr-Hau Lung, Kuo-Wei Ho, Tzu-Chen Yen, Sheng-Chieh Chan, Shu-Tian Chen, Ming-Feng Tsai, Ming-Szu Hung

**Affiliations:** 1Department of Nuclear Medicine, Jen-Ai Hospital, Dali Branch, No. 483, Dong Rong Rd., Dali District, Taichung City 412224, Taiwan; t11815@ms.sltung.com.tw; 2Department of Respiratory Care, Chang Gung University of Science and Technology, No. 2, Sec. West, Jiapu Rd., Puzi City 613, Chiayi County, Taiwan; sugarcan99@cgmh.org.tw; 3Department of Pulmonary and Critical Care Medicine, Chang Gung Memorial Hospital, Chiayi Branch, No. 6, Sec. West, Jiapu Rd., Puzi City 613, Chiayi County, Taiwan; chestmed@adm.cgmh.org.tw; 4Department of Radiation Oncology, Kaohsiung Chang Gung Memorial Hospital, Chang Gung University College of Medicine, No. 123, Dapei Rd., Niaosong Dist., Kaohsiung City 833, Taiwan; yjhuang@adm.cgmh.org.tw; 5Department of Medical Research, Chang Gung Memorial Hospital, Chiayi Branch, No. 6, Sec. West, Jiapu Rd., Puzi City 613, Chiayi County, Taiwan; jrhaulung@cgmh.org.tw; 6Department of Nuclear Medicine, Chang Gung Memorial Hospital, Chiayi Branch, No. 6, Sec. West, Jiapu Rd., Puzi City 613, Chiayi County, Taiwan; kwhomd@adm.cgmh.org.tw (K.-W.H.); tsaimifo@adm.cgmh.org.tw (M.-F.T.); 7Department of Nuclear Medicine and Molecular Imaging Center, Chang Gung Memorial Hospital, Chang Gung University College of Medicine, No. 5, Fuxing St., Guishan Dist., Taoyuan City 333, Taiwan; yen1110@cgmh.org.tw; 8Department of Nuclear Medicine, Hualien Tzu-Chi Hospital, Buddhist Tzu-Chi Medical Foundation, No. 707, Sec. 3, Chung-Yang Rd., Hualien City 970, Taiwan; wi110911@tzuchi.com.tw; 9Department of Medicine, School of Medicine, Tzu Chi University, No. 701, Sec. 3, Chung-Yang Rd., Hualien City 970, Taiwan; 10Department of Diagnostic Radiology, Chang Gung Memorial Hospital, Chiayi Branch, No. 6, Sec. West, Jiapu Rd., Puzi City 613, Chiayi County, Taiwan; 11Department of Medicine, College of Medicine, Chang Gung University, No. 259, Wenhua 1st Rd., Guishan Dist., Taoyuan City 333, Taiwan

**Keywords:** ^18^F-FDG PET, adenocarcinoma of lung, tyrosine kinase inhibitors, early response evaluation, survival

## Abstract

**Simple Summary:**

Epithelial growth factor receptor-tyrosine kinase inhibitors (EGFR-TKIs) are the first-line therapy for patients with advanced-stage lung adenocarcinoma with EGFR mutations. However, 17–31% of these patients do not respond to therapy, making early evaluation of treatment response crucial. This prospective study investigates the value of ^18^F-fluorodeoxyglucose positron emission tomography/computed tomography (^18^F-FDG PET/CT) for timely prediction of response and survival of these patients. We evaluated 30 patients with stage IIIB/IV lung adenocarcinomas and EGFR mutations, receiving first-line EGFR-TKI therapy. ^18^F-FDG PET/CT was performed before and two weeks after initiation of treatment. Positron Emission Tomography Response Criteria in Solid Tumors served as an independent predictor of non-progressive disease; baseline and change of metabolic tumor volume represented independent predictors of progression-free survival and overall survival, respectively. Therefore, ^18^F-FDG PET/CT is an early predictor of outcomes and individual prognosis of patients with stage IIIB/IV lung adenocarcinomas and EGFR mutations receiving first-line EGFR-TKI therapy.

**Abstract:**

This study aims to investigate the role of ^18^F-fluorodeoxyglucose positron emission tomography/computed tomography (^18^F-FDG PET/CT) in early prediction of response and survival following epithelial growth factor receptor (EGFR)–tyrosine kinase inhibitor (TKI) therapy in patients with advanced lung adenocarcinomas and EGFR mutations. Thirty patients with stage IIIB/IV lung adenocarcinomas and EGFR mutations receiving first-line EGFR-TKIs were prospectively evaluated between November 2012 and May 2015. EGFR mutations were quantified by delta cycle threshold (dCt). ^18^F-FDG PET/CT was performed before and 2 weeks after treatment initiation. PET response was assessed based on PET Response Criteria in Solid Tumors (PERCIST). Baseline and percentage changes in the summed standardized uptake value, metabolic tumor volume (bsumMTV and ΔsumMTV, respectively), and total lesion glycolysis of ≤5 target lesions/patient were calculated. The association between parameters (clinical and PET) and non-progression disease after 3 months of treatment in CT based on the Response Evaluation Criteria in Solid Tumors Version 1.1 (nPD_3mo_), progression-free survival (PFS), and overall survival (OS) were tested. The median follow-up time was 19.6 months. The median PFS and OS were 12.0 and 25.3 months, respectively. The PERCIST criteria was an independent predictor of nPD_3mo_ (*p* = 0.009), dCt (*p* = 0.014) and bsumMTV (*p* = 0.014) were independent predictors of PFS, and dCt (*p* = 0.014) and ΔsumMTV (*p* = 0.005) were independent predictors of OS. ^18^F-FDG PET/CT achieved early prediction of outcomes in patients with advanced lung adenocarcinomas and EGFR mutations receiving EGFR-TKIs.

## 1. Introduction

Epithelial growth factor receptor (EGFR) belongs to a family of receptor tyrosine kinases that can trigger a series of signaling pathways leading to cell growth, proliferation, and survival [1,2]. A subset of lung adenocarcinomas has driver mutations in the EGFR gene, in which tumor cell survival is sensitively dependent on the EGFR pathway signaling [3]. Tyrosine kinase inhibitors (TKIs), which block the tyrosine kinase domain of EGFR, can inhibit mutant EGFR and thus subsequent downstream pathways [4,5]. Previous randomized trials have shown that patients with advanced-stage lung adenocarcinoma and mutations in EGFR, exhibit improved overall response rates, progression-free survival (PFS) (gefitinib: 24.9% vs. carboplatin/paclitaxel: 6.7%; *p* < 0.001), and quality of life following EGFR-TKI therapy compared to traditional chemotherapy [6,7]. Hence, EGFR-TKIs have been established as the first-line therapy for these patients [8].

Despite the increased response rates of EGFR-TKIs relative to those of chemotherapy in lung adenocarcinomas with mutant EGFR, 17–31% of patients do not respond to therapy [9,10,11,12]. This may be due to the heterogeneity of mutations within the tumor, resulting from cancer cells with different genetic alterations in tumor tissue [13,14]. Mutation analysis from only biopsy specimens is suboptimal for predicting response [15]. Intratumoral heterogeneity of EGFR mutations is reported as a potential source of EGFR-TKIs treatment failure [16]. Therefore, an early evaluation of the treatment response remains important. However, conventional computed tomography (CT) imaging has limitations in this respect. Since EGFR-TKIs mostly have a cytostatic effect, as opposed to a cytotoxic effect, tumors may not regress on CT imaging, despite effective treatment [17].

^18^F-fluorodeoxyglucose positron emission tomography/CT (^18^F-FDG PET/CT) is a proven staging modality in patients with non–small-cell lung cancer (NSCLC) [18,19] and is based on high glucose metabolism in tumor cells that show increased glucose transport protein expression and hexokinase activity [20]. In addition to staging, it is increasingly used to assess tumor response and outcomes [21]. Previous studies have demonstrated that the change in ^18^F-FDG PET or PET/CT after 1 to 2 weeks of EGFR-TKI treatment can predict a conventional CT response at 2 to 3 months in patients with lung cancer [22,23,24,25,26]. However, because the ^18^F-FDG PET analytical methods are diverse in terms of lesion selection (such as single or multiple), the definition of the volume of interest (VOI), selected parameters (such as standardized uptake value [SUV] or total lesion glycolysis [TLG]), and response criteria [22,23,24,25,26,27,28,29], there is lack of uniform consensus of the use of ^18^F-FDG PET/CT in patients with lung cancer treated with EGFR-TKIs. The role of ^18^F-FDG PET/CT for improving the management strategy remains unclear.

This prospective study aimed to investigate the value of evaluating ^18^F-FDG PET/CT at baseline and 2 weeks after the initiation of treatment for the early prediction of response and survival in patients with advanced lung adenocarcinomas and EGFR mutations receiving first-line EGFR-TKIs.

## 2. Materials & Methods

### 2.1. Patients

We prospectively enrolled all patients with stage IIIB or IV lung adenocarcinoma with EGFR mutations who met the eligibility criteria and received EGFR-TKIs as first-line monotherapy between November 2012 and May 2015, and for whom ^18^F-FDG PET/CT data were available. Patients were excluded if they had a known history of malignancy, including prior lung cancer, prior surgery or systemic therapy for lung cancer, or symptomatic brain metastasis. The TNM stage was determined by chest and upper abdomen CT including adrenals with and without contrast, brain magnetic resonance imaging (MRI) and bone scan based on the 7th edition of the American Joint Committee on Cancer [30]. We also included ^18^F-FDG PET/CT in staging for the enrolled patients. The study was approved by the institutional review board of Chang Gung Memorial Hospital (approval number: 101-3153C, dated 20 September 2012) and was performed in accordance with the principles of the Declaration of Helsinki. All participants provided written informed consent.

All patients underwent EGFR mutation diagnostics. Lung-cancer tissue samples were obtained by bronchoscopic, CT-guided or surgical biopsy, and were confirmed by pathologists. Biopsies were performed before the first FDG PET/CT examinations. Genomic DNA of the tumor samples was extracted from formalin-fixed paraffin-embedded (FFPE) sections using a QIAamp DNA FFPE Tissue Kit (Qiagen, Hilden, Germany) following the manufacturer’s protocol. We performed the EGFR mutation analysis using the Therascreen EGFR RGQ PCR kit (Qiagen) based on Amplification Refractory Mutation System technology. For each EGFR mutation-positive signal, subtracting the control assay cycle threshold (Ct) from the mutation assay Ct gave the delta Ct (dCt). The dCt was inversely proportional to the logarithmic values of the percentage of mutant EGFR DNA, in accordance with our previous study [31]. This means that the higher the dCt value, the lower the amount of mutant EGFR DNA content. In this study, we used the dCt to indicate the relative abundance of the EGFR mutant allele.

Patients were treated with gefitinib (250 mg/d), erlotonib (150 mg/d), or afatinib (40 mg/d) until progressive disease (PD). ^18^F-FDG PET/CT was performed before (baseline) and 2 weeks after initiation of EGFR-TKI treatment. Diagnostic chest CT scans were performed at both baseline and every 3 months after treatment until PD.

### 2.2. ^18^F-FDG PET/CT Image Acquisition

All patients fasted for ≥6 h before undergoing PET/CT examinations. In addition, we tested serum glucose levels via finger-stick sampling before intravenous administration of 5.2 MBq/kg (0.14 mCi/kg) of ^18^F-FDG to ensure they were <150 mg/dL. The ^18^F-FDG PET images were obtained using a Biograph TruePoint 64 PET/CT scanner (Siemens, Knoxville, TN, USA) equipped with lutetium oxyorthosilicate detectors and a 64-slice CT scanner. The scanner permits simultaneous acquisition of 55 transverse planes of 3.0-mm thickness that encompass a 70-cm axial field of view. A low-dose unenhanced CT scan using a standard protocol of 120 kV, 50 mA, tube rotation time of 0.5 s per rotation, and a pitch of 0.8 was performed first for PET attenuation correction. About 60 min after injection of the ^18^F-FDG, a whole-body PET scan was acquired over 30–40 min (3 min/bed) in two-dimensional mode starting at the feet and proceeding to the head. The PET images were then reconstructed using ordered subsets expectation maximization iterative algorithms with 14 subsets, 2 iterations, and 168 × 168 pixels. In-plane resolution of 3.0 mm and axial resolution of 4.1 mm were obtained.

### 2.3. ^18^F-FDG PET/CT Image Analysis

A single researcher (Huang YE) performed the ^18^F-FDG PET/CT analysis. According to the Positron Emission Tomography Response Criteria in Solid Tumors (PERCIST) 1.0 recommendations [32], we used the SUV normalized to lean body mass (SUL), which was considered to have better patient-to-patient consistency for analysis in this study. The maximum and mean SUL (SUL_max_ and SUL_mean_, respectively) within a VOI were calculated as follows:(SUL_max_ = max. tissue activity concentration [MBq/g]/(injected dose [MBq]/lean body mass [g]).
SUL_mean_ = mean tissue activity concentration [MBq/g]/(injected dose [MBq]/lean body mass [g]). 

We determined normal background ^18^F-FDG activity in the inferior right lobe of the liver, consisting of the mean SUL and standard deviation in a 3-cm-diameter spherical region of interest (ROI). If the liver showed metastasis, the 3-cm-diameter spherical ROI was drawn in the region without obvious metastatic lesions. Measurable tumor lesions were those with a SUL_max_ ≥1.5-fold higher than the liver SUL_mean_ plus two times its standard deviation in the baseline study.

The VOIs of the selected tumor lesions were determined using a threshold-based method. The lower bounds of the VOIs were defined as two standard deviations above the liver SUL_mean_. The values of SUL_max_, SUL_mean_, and metabolic tumor volume (MTV) within the tumor lesion VOIs were measured. We calculated the TLG as SUL_mean_ × MTV.

### 2.4. Response Assessment Based on ^18^F-FDG PET/CT

On the basis of the PERCIST 1.0 criteria, we selected the measurably hottest (high^18^F-FDG uptake) single-tumor lesion at baseline and the hottest single-tumor lesion on the post-treatment scan. The lesions were not required to be the same between the two scans. The SUL_max_ values of the hottest single-tumor lesions were measured, and the percentage of changes in SUL_max_ (ΔSUL) from that in the baseline PET scans to that in the post-treatment PET scans was calculated. Complete metabolic response (CMR) was defined as complete resolution of tumor ^18^F-FDG uptake; partial metabolic response (PMR) was defined as ΔSUL ≤ −30%, and progressive metabolic disease (PMD) was defined as either ΔSUL > 30% or development of a new lesion. Finally, stable metabolic disease (SMD) was defined as the absence of CMR, PMR, and PMD. The metabolic responders (MRs) included patients with CMR and PMR, whereas the metabolic nonresponders (nMRs) included patients with SMD and PMD.

For exploratory analysis, we included as many as five measurably hottest target tumor lesions and no more than two per organ as target lesions at baseline scan, and we measured the same five lesions on the post-treatment scan. The SUL_max_, MTV, and TLG of the target lesions were measured. The percentage changes in the sums of SUL_max_, MTV, and TLG of the target lesions from the baseline to post-treatment scans were calculated and represented by ΔsumSUL, ΔsumMTV, and ΔsumTLG, respectively. We also included baseline SUL_max_ of the hottest single-tumor lesions (bSUL) and baseline sums of SUL_max_ (bsumSUL), MTV (bsumMTV), and TLG (bsumTLG) of the target lesions.

### 2.5. Response Assessment Based on CT

In an independent radiologic review blinded to PET/CT results, the response to EGFR-TKIs was assessed essentially by comparing the chest and upper abdomen CT images including adrenals with and without contrast obtained 3 months after treatment with the baseline scans analyzed by a consensus between two readers (Hung MS and Chen ST). The decision to use contrast was at the discretion of the clinical physician, based on patients’ clinical condition (such as allergy, renal insufficiency or other contraindications). Equivalent CT images at baseline and on 3 months were used for comparison. In addition to the chest CT images, brain MRI or bone scan data were also included for the distant metastasis survey. We identified target lesions (≤two lesions per organ, maximum of five lesions). The tumor response was classified as complete response (CR), partial response (PR), stable disease (SD), or PD according to the Response Evaluation Criteria in Solid Tumors Version 1.1 (RECIST 1.1) [33]. We dichotomized the patients into PD versus non-PD (CR + PR + SD) groups, as these criteria serve as the basis for determining whether to continue or discontinue EGFR-TKI treatment. The disease control rate was defined as the percentage of non-PD patients.

### 2.6. Statistical Analysis

The main outcome measures were non-PD after 3 months of EGFR-TKI treatment in CT based on RECIST 1.1 (nPD_3mo_), PFS, and overall survival (OS). PFS was calculated from the initiation of EGFR-TKIs to PD. OS was calculated from the start of EGFR-TKIs to death or the last follow-up date. We used univariate and multivariate analysis to evaluate the predictive abilities of clinical [age, sex, smoking, initial serum carcinoembryonic antigen (CEA) level, classical EGFR mutations, and dCt] and PET (PERCIST, ΔsumSUL, ΔsumMTV, and ΔsumTLG, ΔsumTLG, bSUL, bsumSUL, bsumMTV and bsumTLG) parameters with respect to nPD_3mo_, PFS, and OS. The correlation between dCt and baseline PET parameters was evaluated using Spearman’s correlation analysis.

For the prediction of nPD_3mo_, we used the Mann–Whitney U and chi-square tests to compare continuous and categorical variables, respectively, in univariate analysis. Logistic regression was used for the subsequent multivariate analysis. If an independent predictor for nPD_3mo_ was identified, the sensitivity, specificity, positive predictive value (PPV), negative predictive value (NPV), and accuracy of it for predicting nPD_3mo_ was calculated directly for categorical variables and based on receiver-operating characteristic curves for continuous variables.

For PFS and OS analysis, survival curves were plotted using the Kaplan–Meier method and compared using the log-rank test in the univariate analysis. The most optimal cutoff value started from the median and increased and decreased iteratively, until it reached the most significant *p* value in log-rank test. Cox regression was used in the multivariate analysis.

To restrict the number of variables in the multivariate analysis, we chose parameters that were found to be significant by univariate analysis, in addition to dCt, PERCIST, ΔsumSUL and ΔsumTLG, which have been previously shown to be prognostic factors for patients with NSCLC treated with EGFR-TKIs as variables [22,23,24,25,31]. A two-tailed *p* value < 0.05 was considered to indicate a significant difference.

## 3. Results

### 3.1. Patient Characteristics

We enrolled 34 patients with stage IIIB or IV lung adenocarcinomas and EGFR mutations, receiving EGFR-TKIs as a first-line therapy and had available ^18^F-FDG PET/CT. A patient flow diagram is shown in Figure 1. Two patients did not receive EGFR-TKI treatment and were lost during follow-up, and one patient died due to severe sepsis within 2 weeks of EGFR-TKI treatment. Therefore, these three patients could not be evaluated for treatment response and were excluded. One patient who received thoracic spine radiation therapy was also excluded. Consequently, 30 patients (18 women, 12 men; age range, 40–91 years; median age, 71 years) were included in the final analyses (Table 1). Twenty patients received gefitinib, seven received erlotinib, and three received afatinib. All 30 patients underwent baseline diagnostic chest CT, baseline and 2-week ^18^F-FDG PET/CT, and 3-month follow-up diagnostic chest CT. The median follow-up time was 19.6 months (range, 2.97–33.0 months), with 15 survivors at the end of the follow-up.

### 3.2. Treatment Response Evaluation

Based on the CT response according to the RECIST 1.1 criteria, after 3 months of treatment, there were 6 PD and 24 non-PD (including 23 PR and 1 SD) patients. The disease control rate was 80% (24/30). ^18^F-FDG PET was performed 0–13 days (mean, 3 days) before treatment initiation and 13–16 days (mean, 14 days) after treatment initiation. In the univariate analysis, the MR (PERCIST; *p* = 0.005), lower ΔsumSUL (*p* = 0.004), ΔsumMTV (*p* = 0.005), ΔsumTLG (*p* = 0.003), bsumMTV (*p* = 0.015) and bsumTLG (*p* = 0.013) values were significantly associated with a higher chance of nPD_3mo_, whereas age (*p* = 0.462), sex (*p* = 1.000), smoking (*p* = 1.000), initial serum CEA level (*p* = 0.937), classical EGFR mutations (*p* = 0.254), dCt (*p* = 0.158), bSUL (*p* = 0.251) and bsumSUL (*p* = 0.129) were not. In the multivariate analysis (Table 2), the MR (PERCIST) was the only independent predictor of nPD_3mo_ (*p* = 0.009; odds ratio, 25.0; 95% confidence interval (CI), 2.27–276). The sensitivity, specificity, PPV, NPV, and accuracy of MRs (PERCIST) for predicting nPD_3mo_ were 83% (20/24), 83% (5/6), 95% (20/21), 56% (5/9), and 83% (25/30), respectively. The PPV of 95% indicated that a very high percentage of the MRs (based on PERCIST criteria) showed nonprogression on CT at 3 months. The NPV of 56% indicated that more than half of the nMRs revealed progression on CT at 3 months, despite the presence of an activating EGFR mutation. Representative images of ^18^F-FDG PET responses during EGFR-TKI treatment are shown (Figure 2).

### 3.3. PFS Evaluation

The median PFS was 12.0 months (95% CI: 10.9–13.1 months). As shown in the univariate analysis, the dCt (*p* = 0.001), ΔsumSUL (*p* = 0.010), bsumMTV (*p* = 0.030) and bsum TLG (*p* = 0.024) were significantly associated with PFS. The PERCIST criteria (*p* = 0.059) showed marginally significant association with PFS. On the other hand, age (*p* = 0.239), sex (*p* = 0.621), smoking (*p* = 0.220), initial serum CEA level (*p* = 0.142), classical EGFR mutations (*p* = 0.183), ΔsumMTV (*p* = 0.149), ΔsumTLG (*p* = 0.085), bSUL (*p* = 0.126) and bsumSUL (*p* = 0.488) were not associated with PFS. Multivariate analysis (Table 3) revealed that dCt (*p* = 0.014; hazard ratio [HR], 4.85; 95% CI, 1.38–17.1) and bsumMTV (*p* = 0.014; HR, 5.60; 95% CI, 1.43–22.0) were independent predictors of PFS. Patients with dCt <6 (14.3 months; 95% CI, 9.22–19.4 months vs. 2.43 months; 95% CI, 0.01–4.86 months; *p* = 0.001) and bsumMTV <40 cm^3^ (19.5 months; 95% CI, 9.07–30.0 months vs. 8.97 months; 95% CI, 1.58–16.4 months; *p* = 0.030) showed improved PFS (Figure 3A,B).

### 3.4. OS Evaluation

The median OS was 25.3 months (95% CI: 20.1–30.5 months). As per the univariate analysis, only ΔsumMTV (*p* = 0.038) was significantly associated with OS. In the multivariate analysis, dCt (*p* = 0.014; HR, 9.84; 95% CI, 1.58–61.2) and ΔsumMTV (*p* = 0.005; HR, 13.1; 95% CI, 2.15–79.4) were independent predictors of OS (Table 4). Patients with dCt < 6 demonstrated better OS, although the difference was not significant (25.3 months; 95% CI, 20.9–29.8 months vs. 12.8 months; 95% CI, 0–28.1 months; *p* = 0.265). Patients with ΔsumMTV < −60% demonstrated significantly longer OS (30.9 months; 95% CI, 22.4–39.4 months vs. 20.1 months; 95% CI, 10.3–29.9 months; *p* = 0.038; Figure 3C).

### 3.5. Correlation Analyses

The bsumMTV (Spearman’s r: 0.366, *p* = 0.043) and bsumTLG (Spearman’s r: 0.361, *p* = 0.046) were found to be significantly associated with dCt, whereas bSUV (Spearman’s r: −0.115, *p* = 0.538) and bsumSUV (Spearman’s r: 0.150, *p* = 0.422) were not.

## 4. Discussion

This study assessed the usefulness of ^18^F-FDG PET/CT for the early prediction of outcome in patients with advanced lung adenocarcinomas and EGFR mutations receiving first-line EGFR-TKI therapy. Based on the change in ^18^F-FDG PET/CT after 2 weeks of therapy, the PERCIST comprised an independent predictor of nonprogression after 3 months of therapy, and ΔsumMTV was an independent predictor of OS.

Zander et al. [22] reported that a >30% reduction in the SUV of the single hottest lesion, as recommended by the PERCIST criteria, after 1 week of erlotinib treatment could predict nonprogression on CT at 6 weeks. A similar finding based on the change in the summed SUVs of target lesions has been reported [23]. In addition, the change in the summed TLG of the target lesions after 1–2 weeks of EGFR-TKI therapy was reported to predict CT response at 2–3 months [24,25]. Together, these studies indicated that an early change in ^18^F-FDG PET can predict the response on a later conventional CT scan in patients with NSCLC treated with EGFR-TKIs. Changes in cellular metabolism usually precede structural changes in tumor cells after treatment [21]. This is the case especially in tumors treated with cytostatic agents rather than cytotoxic agents [26,34]. In EGFR-TKI-sensitive tumor cells, the downregulation of glucose uptake via glucose transporter translocation and hexokinase activity reduction was observed within 48 h after TKI treatment in a mouse model [35]. Consistent with these reports, our study revealed that the PERCIST criteria based on PET at 2 weeks was an independent predictor of nonprogression on CT at 3 months. In this study, 95% of the MRs (based PERCIST criteria) showed nonprogression on CT at 3 months, which offered a very high level of confidence for continuing the therapy. On the other hand, 56% of the nMRs showed progression on CT at 3 months, thus, closer monitoring is warranted for this group of patients. ^18^F-FDG PET/CT based on the PERCIST criteria could aid in adjusting the management strategy during early EGFR-TKI therapy for patients with advanced lung adenocarcinomas and EGFR mutations.

Previous studies showed that the correlation of change with treatment response was weaker for SUV or SUL than that for summed TLG in patients with non-small-cell lung cancer who were treated with EGFR-TKIs, possibly because of bone flare [24,25]. These findings differed from our results as our results indicated that the PERCIST criteria was the only independent predictor of nonprogression on CT at 3 months. However, these studies included relatively unselected patient populations regarding histology type, EGFR mutation status, and primary or palliative choice of TKI treatment, in contrast to our study. We speculated that the susceptibility to bone flare among the various PET parameters may differ among patient populations, which might explain the conflicting results.

In previous studies, an early change in ^18^F-FDG PET predicted PFS and OS in patients with NSCLC treated with EGFR-TKIs [22,23,24,36,37]. Reduction in the summed SUVs of the target lesions of >15–30% after 2–14 days of EGFR-TKI treatment was associated with significantly longer PFS and OS [23,36,37]. Similar results were obtained with the application of PERCIST or TLG [22,24]. Consistent with these reports, our results demonstrated a significant association between a reduction in ΔsumSUL of >40% and longer PFS, based on the univariate analysis. Notably, we found that bsumMTV and ΔsumMTV were independent predictors of PFS and OS, respectively. MTV measures the volumes of metabolically active tumors and thus incorporates tumor volume and metabolic activity as TLG. Therefore, bsumMTV and ΔsumMTV indicate baseline and the change in metabolic tumor burden, respectively. Although the prognostic significance of MTV has been shown in various malignancies, including lung cancer [38,39,40,41,42], it has only been evaluated in a few studies for patients with NSCLC treated with EGFR-TKIs and the results were controversial [29,43]. Hong IK et al. [43] showed that a high pre-treatment MTV was independently associated with shorter PFS and OS, whereas Cook et al. [29] found no significant association between MTV change and OS. The two studies measured only the primary tumor, therefore, the biological heterogeneity between the primary tumor and its metastatic progeny [44] was not considered. In our study, we measured the MTV of up to five target lesions and found that bsumMTV and ΔsumMTV had independent prognostic value. A bsumMTV < 40 cm^3^ was significantly associated with improved PFS, and a > 60% reduction in ΔsumMTV was significantly associated with improved OS. We infer that tumor burden assessed by MTV yields important prognostic information in patients with advanced lung adenocarcinomas and EGFR mutations receiving first-line EGFR-TKI therapy.

In this study, the dCt was an independent predictor of PFS and OS. The PFS of patients with dCt < 6 was significantly longer than that of patients with dCt ≥ 6. A lower dCt indicates higher content of mutant EGFR DNA in lung cancer tissue. In previous studies [31,45], a higher percentage of mutant EGFR DNA was associated with a longer PFS after EGFR-TKI treatment, probably because of increased EGFR gene copy number. This could explain the prognostic value of dCt in this study. Meanwhile, we found dCT to be significantly related to bsumMTV and bsumTLG. A similar finding was recently reported showing that EGFR activating mutation allele frequency was significantly related to baseline TLG [46]. Therefore, we suggest that dCt is an important prognostic biomarker and is correlated with baseline tumor burden in patients with lung cancer who receive treatment with EGFR-TKIs.

Our study has several limitations. First, the sample size was relatively small and the actual performance of the PET parameters that did not show statistical significance in this study should be further evaluated in a larger patient population. Second, we used SUL_max_ instead of SUL_peak_ as suggested by the PERCIST 1.0 criteria. SUL_max_ measures single-voxel SUL, whereas SUL_peak_ assesses SUL in a sphere with a diameter of approximately 1.2 cm. SUL_max_ is commonly used but reportedly less reproducible than SUL_peak_ [32]. Further studies are required to compare the two parameters to see if the results change. Third, a mathematical feature analysis of ^18^F-FDG distribution heterogeneity was not performed in this study. There is evidence that the measurement of change in intratumoral heterogeneity by ^18^F-FDG PET using feature analysis is independently associated with OS and treatment response in patients with NSCLC treated with EGFR-TKIs [29]. Further investigation is required to include these parameters and search for the optimal imaging biomarker. Finally, this study did not include the third-generation EGFR-TKIs (osimertinib) since the first and second-generation EGFR-TKIs were standard treatment at the time of the study. With the increasing use of osimertinib as first-line therapy, use of ^18^F-FDG PET/CT parameters for evaluating the response of patients with NSCLC and EGFR mutations to osimertinib warrants further investigation.

## 5. Conclusions

In patients with advanced lung adenocarcinomas and EGFR mutations treated with first-line EGFR-TKIs, ^18^F-FDG PET/CT predicted outcomes early and provided individual prognosis. The PERCIST criteria comprised an independent predictor of non-PD, baseline MTV was an independent predictor of PFS, and change of MTV was an independent predictor of OS. EGFR gene mutation quantification assessed by dCt was independently predictive of PFS and OS.

## Figures and Tables

**Figure 1 cancers-14-01507-f001:**
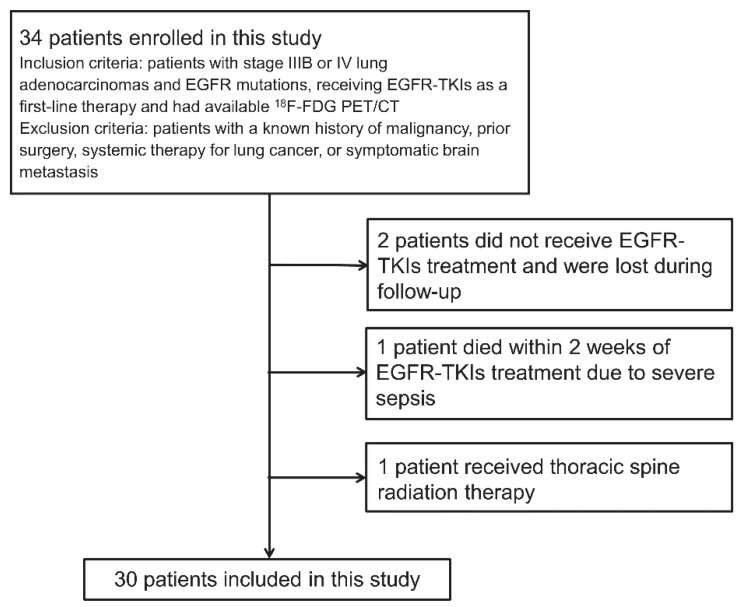
Patient flow diagram.

**Figure 2 cancers-14-01507-f002:**
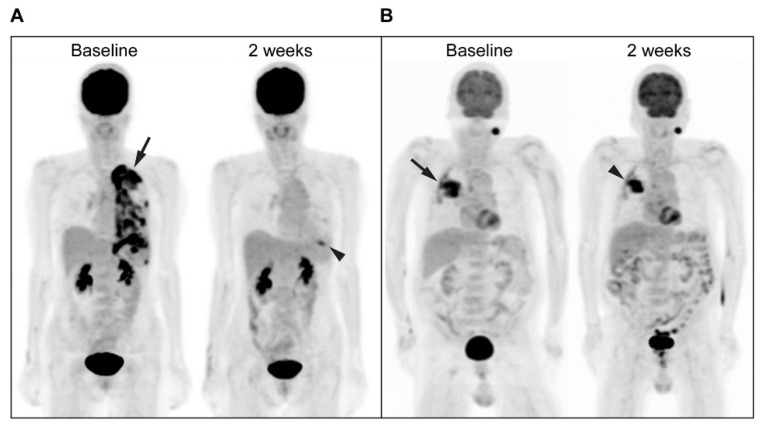
Illustrative images of ^18^F-FDG PET responses. (**A**) Maximum-intensity projection (MIP) ^18^F-FDG PET images at baseline and after 2 weeks of gefitinib treatment of a 58-year-old woman with stage IV lung adenocarcinoma and mutant EGFR (exon 19 del) showing marked ^18^F-FDG response visually. The SUL_max_ of the single hottest lesion decreased from 10.7 (left upper pleura, arrow) to 4.10 (lingula of left lung, arrowhead), with a ΔSUL of −61.5% and a PMR based on PERCIST 1.0. She achieved PR at 3 months. (**B**) MIP ^18^F-FDG PET images at baseline and after 2 weeks of gefitinib treatment of a 78-year-old man with stage IV lung adenocarcinoma and mutant EGFR (L858R) showing SMD (nMR) based on PERCIST 1.0. The SUL_max_ of the single hottest lesion decreased from 9.37 (right upper lobe pulmonary mass, arrow) to 8.91 (the same lesion, arrowhead), with a ΔSUL of −4.90%. There was also an incidental left parotid tumor. He had PD (a new left adrenal metastasis) at 3 months.

**Figure 3 cancers-14-01507-f003:**
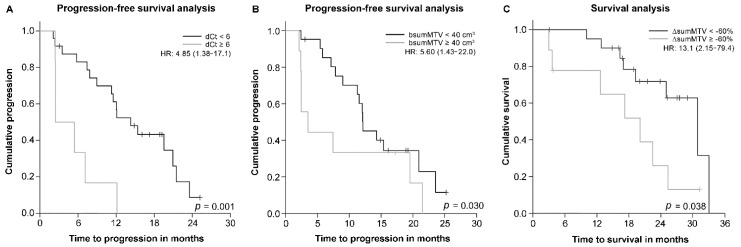
Kaplan–Meier survival curve showing differences in progression-free survival between patients with a dCt of ≥6 and those with a dCt of <6 (**A**) and between patients with a bsumMTV ≥ 40 cm^3^ and < 40 cm^3^ (**B**). (**C**) Kaplan–Meier survival curve showing differences in overall survival between patients with a ΔsumMTV of ≥−60% and those with a ΔsumMTV of <−60%. HR, hazard ratio.

**Table 1 cancers-14-01507-t001:** Patient characteristics.

Characteristics	No. (%)
Number of patients	30 (100)
Age (y)	
Median	71
Range	40–91
Sex	
Female	18 (60)
Male	12 (40)
Smoking	
Never	22 (73)
Ever	8 (27)
ECOG performance status	
0	1 (3)
1	29 (97)
AJCC clinical stage	
IIIB	1 (3)
IV	29 (97)
EGFR mutation type	
Classical	25 (83)
Exon 21 L858R	13 (43)
Exon 19 deletion	11 (37)
Both	1 (3)
Others	5 (17)
CEA (ng/mL)	
Median	6.92
Range	0.5–1034
CT response at 3 months (RECIST)	
PR	23 (77)
SD	1 (3)
PD	6 (20)
PET response at 2 weeks (PERCIST)	
PMR	21 (70)
SMD	9 (30)

ECOG, Eastern Cooperative Oncology Group; AJCC, American Joint Committee on Cancer; EGFR, epithelial growth factor receptor; RECIST, Response Evaluation Criteria in Solid Tumors; PR, partial response; SD, stable disease; PD, progression disease; PERCIST, Positron Emission Tomography Response Criteria in Solid Tumors; PMR, metabolic partial response; SMD, metabolic stable disease.

**Table 2 cancers-14-01507-t002:** Multivariate analysis of parameters and association with nPD_3mo_ (*n* = 30).

Parameter	Nonprogression (*n* = 24)	Progression (*n* = 6)	*p* Value	Odds Ratio (95% CI)
dCt	3.65	5.22	0.180	0.66 (0.41–1.07)
	(8.54–1.07)	(8.89–2.63)		
MR (PERCIST)	83	17	0.009 *	25.0 (2.27–276)

ΔsumSUL (%)	−46.2	−20.7	0.516	0.003 (0–0.41)
	(−72.5 to 24.4)	(−49.7 to −3.24)		
ΔsumMTV (%)	−77.0	−35.8	0.191	0.03 (0.001–0.74)
	(−99.8 to 5.35)	(−65.4 to 82.5)		
ΔsumTLG (%)	−80.8	−39.6	0.272	0.28 (0.001–0.79)
	(−99.8 to 2.47)	(−72.3 to 76.7)		
bsumMTV (cm^3^)	63.00	80.60	0.080	0.99 (0.98–1.00)
	(1.090–287.2)	(8.410–287.3)		
bsumTLG (g)	247.9	307.8	0.085	1.00 (0.99–1.00)
	(2.950–1124)	(24.47–1124)		

MR, metabolic responder; PERCIST, Positron Emission Tomography Response Criteria in Solid Tumors; SUL, standardized uptake value normalized to lean body mass; MTV, metabolic tumor volume; TLG, total lesion glycolysis; CI, confidence interval. The sample statistics presented in this table were frequency (percentage, %) for categorical variables and median (range) for continuous variables. * *p* < 0.05 indicates a significant difference.

**Table 3 cancers-14-01507-t003:** Multivariate analysis of parameters and association with PFS (*n* = 30).

Parameter	Median PFS (Months) (95% CI)	*p* Value	Hazard Ratio (95% CI)
dCt		0.014 *	4.85 (1.38–17.1)
≥6	2.43 (0.01–4.86)		
<6	14.3 (9.22–19.4)		
PERCIST		0.882	0.91 (0.26–3.19)
nMR	3.50 (0–10.2)		
MR	12.1 (2.46–7.31)		
ΔsumSUL		0.134	2.73 (0.77–10.2)
≥−40%	3.50 (0–12.0)		
<−40%	15.4 (7.74–23.1)		
ΔsumTLG		0.107	3.36 (0.77–14.7)
≥−50%	3.50 (0.76–6.24)		
<−50%	12.1 (8.98–15.3)		
bsumMTV		0.014 *	5.60 (1.43–22.0)
≥40 cm^3^	8.97 (1.58–16.4)		
<40 cm^3^	19.5 (9.07–30.0)		
bsumTLG		0.222	2.02 (0.65–6.24)
≥ 300 g	7.59 (4.14–11.0)		
< 300 g	14.8 (11.5–18.1)		

dCt, delta cycle threshold; PERCIST, Positron Emission Tomography Response Criteria in Solid Tumors; SUL, standardized uptake value normalized to lean body mass; TLG, total lesion glycolysis; PFS, progression-free survival; CI, confidence interval. * *p* < 0.05 indicates a significant difference.

**Table 4 cancers-14-01507-t004:** Multivariate analysis of parameters and association with OS (*n* = 30).

Parameter	Median OS (Months) (95% CI)	*p* Value	Hazard Ratio (95% CI)
dCt		0.014 *	9.84 (1.58–61.2)
≥6	12.8 (0–28.1)		
<6	25.3 (20.9–29.8)		
PERCIST		0.636	0.64 (0.30–7.03)
nMR	20.1 (8.23–32.0)		
MR	25.3 (22.5–28.2)		
ΔsumSUL		0.106	0.14 (0.01–1.53)
≥−40%	22.5 (15.1–29.9)		
<−40%	30.9 (30.9–30.9)		
ΔsumMTV		0.005 *	13.1 (2.15–79.4)
≥−60%	20.1 (10.3–29.9)		
<−60%	30.9 (22.4–39.4)		
ΔsumTLG		0.871	1.23 (0.10–14.8)
≥−50%	22.5 (11.5–33.4)		
<−50%	30.9 (22.4–39.4)		

dCt, delta cycle threshold; PERCIST, Positron Emission Tomography Response Criteria in Solid Tumors; SUL, standardized uptake value normalized to lean body mass; TLG, total lesion glycolysis; OS, overall survival; CI, confidence interval. ** p* < 0.05 indicates a significant difference.

## Data Availability

The datasets generated during and/or analyzed during the current study are available from the corresponding author on reasonable request.

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
