# Peer review of "^18^F-Fluorodeoxyglucose PET/CT for Early Prediction of Outcomes in Patients with Advanced Lung Adenocarcinomas and EGFR Mutations Treated with First-Line EGFR-TKIs"

_cancers, 2022, doi:10.3390/cancers14061507_

Round 1

Reviewer 1 Report

Title: 18F-fluorodeoxyglucose PET/CT for early prediction of out- comes in patients with advanced lung adenocarcinomas and EGFR mutations treated with first-line EGFR-TKIs

General Comments:

This manuscript aimed to test the role of PET/CT in the early prediction of response in patients affected by advanced lung adenocarcinoma with EGRF mutation, treated with TKIs. Thirty patients were prospectively enrolled and underwent baseline and after 2 weeks 18F- FDG PET/CT; also chest CT were performed at baseline and after 3 months of therapy and used to assess response to therapy according to RECIST 1.1 criteria. PET/CT parameters such as PERCIST, deltasum MTV, SUL and TLG were analyzed. Then a univariate analysis and a multivariate analysis were performed, also by including clinical and pathological data (such as EGFR mutation was quantified by delta cycle threshold, dCt). Response to therapy at three months, overall survival and progression free survival were considered as outcome. Results showed PERCIST criteria as a significant independent predictor of progressive disease at 3 months. As well as dCt results independent predictor of progression-free survival and dCT and delta sum MTV were independent predictors of overall survival. The topic of the study is very interesting even if the population sample is reduced and the manuscript lacks important details in some sections. Please see specific comments for implementations in the sections.

Specific comments:

Title: ok.

Abstract:

Please, add at the beginning “our study or this study aims to” and then the purpose of the study already present.

Please, specify how nPD3mo was evaluated.

Please, specify the total follow-up period and the enrollment period.

Keywords: ok.

Introduction:

  • The sentence related to Asian non-smoking women seems disconnected from the other sentences, please modify or delete.
  • Please, specify PFS of previous trials with TKIs therapy vs standard chemotherapy.

Material and Methods:

  • Please rewrite in a more schematic way inclusion/exclusion criteria and add all the inclusion criteria (e.g. PET/CT examinations availability).
  • Please, specify with reference the TNM used for the stage assessment and how it was determined (CT? PET/CT?).
  • Please, clearly describe how the biologic samples were obtained and when also related to the first PET/CT examinations.
  • Please, specify how was assessed progressive disease; only chest-CT seems too reductive to assess distant metastasis. In addition, the CTs performed were unenhanced or with contrast media administration? Please, clarify.
  • Please, specify who performed the analysis and if there were a single reader or multiple readers.
  • Please, clarify how RECIST 1.1 criteria were applied only with chest CT examination.
  • Please, justify the following sentence also by adding references: “To restrict the number of variables in the multivariate analysis, we considered only variables that showed some association with outcome measures, including parameters that were close to but did not reach statistical significance, in the univariate analysis.” Statistical significance has a clear meaning and “close to but did not reach statistical significance” sounds too vague and unclear also in consideration of the reduced number of patients enrolled.  

Results:

Generally ok.

Discussion:

Generally ok.

Conclusions:

References:

  • Please add some references as above mentioned.

Tables:

Table 2 layout could be improved.

Figures:

Figure 1: clear and explicative. Figure 2: please check figure description (progree instead of progression).  

Linguistic and typewriting: Slight linguistic improvements are necessary. Please check alto type writing.

Reviewer 2 Report

Thank you for the opportunity to review this paper.  

Huang et al explored the role of 18F-FDG PET scan as a predictor of response in 34 patients with advanced EGFR-mutated NSCLC treated with early-generation EGFR-TKIs.      

Overall, this is a good research paper which strengthen the predictive role of PET scan in EGFR-mutated patients and brings the basis for further analyses. I hope that my comments will improve the value of the paper: 

  • I was wondering if the authors correlated baseline PET scan parameters with response to TKIs; in this occurrence, the three patients who did not receive a post-treatment PET scan could have been included in the analyses 
  • Did the author explore a putative correlation between baseline EGFR mutation dCT and PET parameters? 
  • The authors reported “The most optimal cutoff value started from the median and increased and decreased iteratively, until it reached the most significant P value in log-rank test”; it could be more indicated to calculate FDG-PET parameters using ROC curve analysis, similarly a paper https://doi.org/10.1016/j.jtho.2021.01.1109.
  • The authors should add the number at risk on KM curves 
  • In the discussion session, the authors stated that ‘95% of the MRs (based PERCIST criteria) showed nonprogression on CT at 3 months, which offered a very high level of confidence for continuing the therapy. On the other hand, more than half (56%) of the nMRs showed progression on CT at 3 months, despite the presence of an activating EGFR mutation’. This should be better clarified within the results. 
  • Genes should be typed in Italics. Moreover, a syntactic revision is recommended. 

Lastly, I would ask the authors where stands the novelty of this study and what could be the clinical implications of this research, considering that osimertinib is the current standard frontline treatment for this subgroup of patients. 

Round 2

Reviewer 1 Report

The authors reviewed the manuscript by changing almost all the aspects requested. However, just one aspect still needs some specification. 

Point 6/Response 6 clarification: How was assessed distant abdominal metastasis? please clarify. 

Author Response

Response to Reviewer 1 Comments

 Point 1: The authors reviewed the manuscript by changing almost all the aspects requested. However, just one aspect still needs some specification. Point 6/Response 6 clarification: How was assessed distant abdominal metastasis? Please clarify.

Response 1: The chest CT for lung cancer included upper abdomen routinely in our institution for distant abdominal metastasis survey. This information has been added in Section 2.1 (the first paragraph) and 2.5.

Reviewer 2 Report

the authors revised the paper according the suggestions and I think that the paper is improved and now acceptable 

Author Response

Response to Reviewer 2 Comments

Point 1: The authors revised the paper according the suggestions and I think that the paper is improved and now acceptable.

Response 1: Thank you very much.